# Tool Wear Prediction Based on a Multi-Scale Convolutional Neural Network with Attention Fusion

Qingqing Huang [1,2], Di Wu [1], Hao Huang [1], Yan Zhang [1,2] and Yan Han [1,2,*]

[1] Key Laboratory of Industrial Internet of Things and Networked Control, Ministry of Education, Chongqing University of Posts and Telecommunications, Chongqing 40065, China
[2] Institute of Industrial Internet, Chongqing University of Posts and Telecommunications, Chongqing 401120, China
* Correspondence: hanyan@cqupt.edu.cn

**Abstract:** Compared with traditional machine learning algorithms, the convolutional neural network (CNN) has an excellent automatic feature learning ability and can complete the nonlinear representation from original data input to output by itself. However, the CNN does not sufficiently mine the tool wear information contained in the multi-sensor data due to disregard of the differences in the contribution of different features when extracting features. In this paper, a tool wear prediction method based on a multi-scale convolutional neural network with attention fusion is proposed, which fuses the tool wear degradation information collected by different types of sensors. In the multi-scale convolution module, convolution kernels with different sizes are used to extract the degradation information of different scales in the wear information, and then the attention fusion module is constructed to fuse the multi-scale feature information. Finally, the mapping between tool wear and multi-sensor data is realized through the feature information obtained by residual connection and full connection layer. By comparing the multi-scale convolutional neural network with different attention mechanisms, the experiments demonstrated the effectiveness and superiority of the proposed method.

**Keywords:** tool wear prediction; multi-scale convolution; attention fusion





## 1. Introduction

The tool is an important part of the machine tool. During the machining process, the tool wear directly affects the quality of the workpiece and increases the consumption of production resources. Therefore, to reduce resource consumption and ensure workpiece quality, it is very important to accurately predict the tool wear state.

Currently, the machine learning-based tool wear prediction method is a commonly used method, which usually consists of three steps: handcrafted feature design, degradation behavior learning, and remaining useful life (RUL) estimation [1–3]. First, the sensitive wear degradation features are extracted from the acquired monitoring data using prior knowledge and expertise [4]. Then, these features are fed into machine learning models, such as random forests, Bayesian networks, and support vector machines, to learn the degradation behavior of tool wear and estimate the RUL [5]. For example, Li et al. [6] extracted the three-domain features of the collected signals, then used the t-SNE algorithm to reduce the dimension of the feature matrix, and then used the XGBoost integrated learning algorithm to estimate the tool wear. Liao et al. [7] proposed a tool wear identification model based on kernel principal component analysis and a support vector machine and used the gray wolf algorithm to select the optimal parameters of the model. Deng et al. [8] proposed a tool wear status determination method combining local mean decomposition and the hidden Markov model for the problem of difficulty in extracting wear-related features from the collected sensor signals. Although these machine learning predictive methods can infer

correlations and causality hidden in the data, they still require a lot of effort in handcrafted feature design.

In recent years, given the advantages of the nonlinear representation ability and automatic feature extraction ability of deep learning models, deep learning has been widely used in the field of tool wear prediction [9]. Compared with traditional machine learning, deep learning can better obtain deep-level information hidden in data by using multiple nonlinear modules to extract tool wear-related features [10,11]. For example, models such as deep belief network [12,13], long short-term memory network [14], and convolutional neural network [15] can learn complex mappings from raw signals, which are useful in tool wear prediction, and obtained better performance. Although these methods have achieved impressive results, they cannot learn the data characteristics of multi-sensor tool wear vibration signals with time series characteristics any better.

To obtain richer tool wear information, a multi-scale convolutional neural network (MSCNN) can extract more data features from tool raw data by extracting features of different types and time scales [16,17]. For example, Xu et al. [18] proposed a multi-scale convolution gated recurrent neural network, which realized the tool state monitoring in the process by processing and learning the original vibration signal. Zhu et al. [19] have effectively improved prediction accuracy through the RUL estimation method of time-frequency representation and MSCNN. Jiang et al. [20] proposed a data-driven method based on a bidirectional long short-term memory network and MSCNN for RUL estimation, and the model achieved good prediction performance. The above methods effectively extract multi-sensor information through the MSCNN, and the model obtains good RUL prediction accuracy. However, the MSCNN only simply splices the extracted multi-scale features, and this will inevitably ignore some important information. In addition, the differences in the contribution of different features to the prediction task are not considered, which will bring adverse effects on the prediction performance. As a result, the accuracy of tool wear prediction by the multi-scale network cannot be further improved.

To address the above issues, this paper proposed a method based on a multi-scale convolutional neural network with attention fusion for tool wear prediction. The method uses pre-processed data features as model input, learns the difference information of different sensor data on the time scale by using multi-scale convolution, and then builds an attention fusion module to focus the network's attention on features extracted in the multi-scale convolution module that are highly correlated with tool wear degradation information, while introducing residual connections to allow the model to retain the original multi-sensor feature information and fuse the key degradation information filtered by attention. Finally, the fully connected layer performs deep representation learning on the extracted tool wear degradation features and then inputs them to the regression layer to complete the tool wear prediction.

## 2. Related Word

### 2.1. Multi-Scale Convolutional Neural Network

In the convolutional network, the high-level abstract features of each dimensional sensor signal are mainly extracted through the convolution layer. The operation of one-dimensional convolution is shown in Formula (1):

$$y_j^l = f\left( \sum_{i \in M_j} x_i^{l-1} \omega_{ij}^l + b_j^l \right),　\quad (1)$$

where $y_j^l$ denotes the $j$th feature mapping of layer $l$, $M_j$ denotes the $j$th convolution region of layer $l - 1$, $x_i^{l-1}$ is the $i$th feature mapping of layer $l - 1$, $\omega_{ij}^l$ denotes the corresponding convolution kernel, $b_j^l$ denotes the corresponding bias vector, and $f(x)$ denotes the activation function of layer $l$. Here, the activation function adopts the rectified linear unit (ReLU).

To speed up model training, a batch normalization layer [21] is adopted to normalize the output of the convolution operation. The pooling layer adopts the maximum pooling operation, and the operation method is shown in Formula (2):

$$Z_i^{l+1}(j) = \max_{(j-1)W+1 \leq t \leq jW} \left\{ r_i^l(t) \right\}, \tag{2}$$

where $Z_i^{l+1}(j)$ denotes the value corresponding to the neuron in layer $l+1$, $w$ is the width of the pooling region, and $r_i^l(t)$ denotes the value of the $t$th neuron in the $i$th feature vector of layer $l$.

To extract richer wear information, multi-scale convolutional networks are used to extract features from multi-sensor data. Multi-scale convolutional networks take advantage of the CNN and can automatically learn good feature representation in the time domain and frequency domain. Its multiple branches can extract feature information at different time scales, which solves the limitation of many previous works that only extract features at a single time scale [16].

In this paper, a multi-scale convolutional module is built as shown in Figure 1 using several convolutional layers to extract the features of each branch. At this stage, the multi-scale convolution module consists of several convolutional networks in parallel, with each branch using a different size convolutional kernel, with the size of the convolutional kernel set to $2n + 1$, where $n$ indicates the number of branches and the convolution of the different branches is independent of each other. The network structure of the multi-scale convolutional module is shown in the figure below.

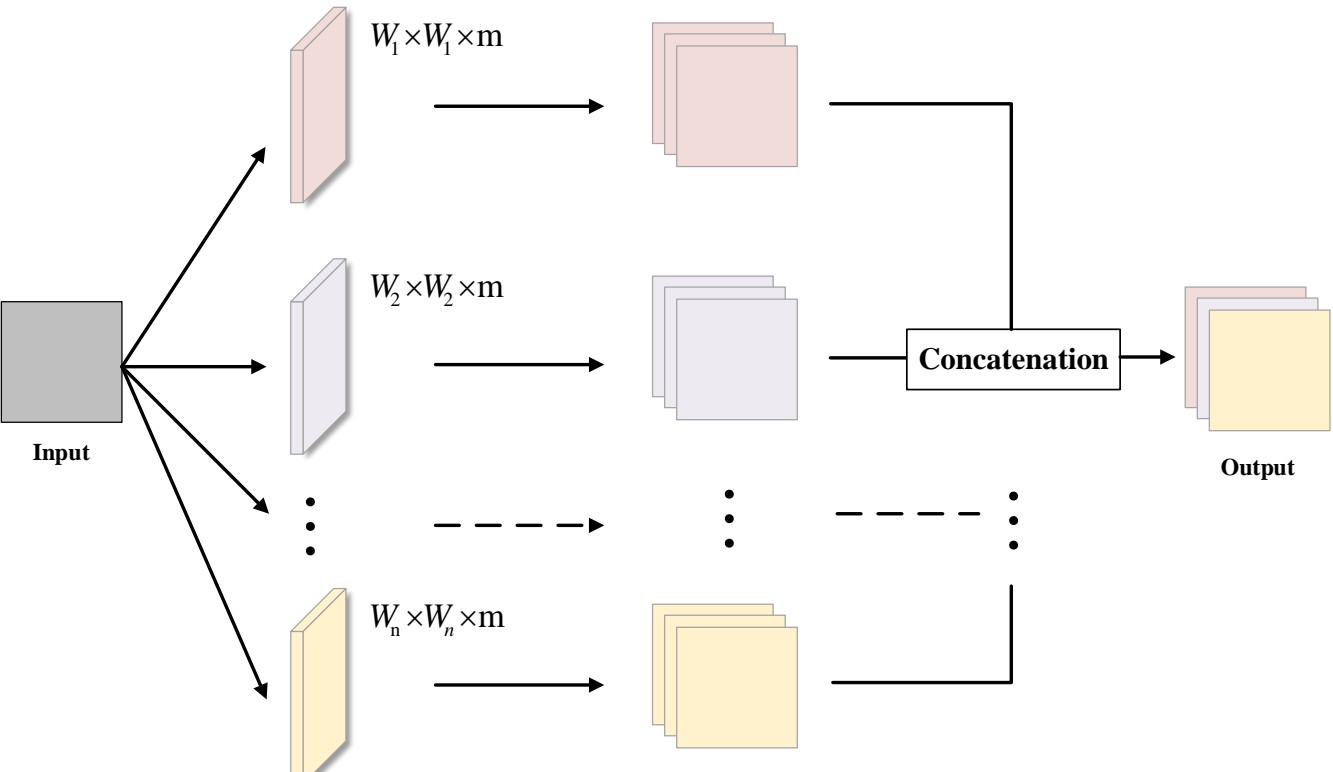

**Figure 1.** The structure of the multi-scale convolutional network.

### 2.2. Attention Mechanism

Attention is inspired by human biological systems that tend to focus on unique parts when processing large amounts of information. It is a means for humans to use limited processing resources to quickly select high-value information from massive amounts of information. Attention mechanisms greatly improve the efficiency and accuracy of perceptual

information processing. Human attention mechanisms can be divided into two categories based on the way they are generated [22]. The first category is bottom-up unconscious attention, driven by external stimuli. The second category is top-down conscious attention, called focused attention. Focused attention is attention that has a predetermined purpose and is dependent on a specific task. It enables humans to consciously and actively focus their attention on an object. Most of the attention mechanisms in deep learning are designed to be task-specific and therefore most of them are focused attention [23].

Inspired by the fact that the human visual system can naturally and efficiently find important regions in complex scenes, attention mechanisms have been introduced into computer vision systems. The basic idea of the attention mechanism in computer vision is for the model to learn to focus on the important regions of an image, concentrating on the important information and ignoring the unimportant information.

In the field of computer vision, the introduction of channel attention into convolutional blocks has allowed networks to show great potential in terms of performance improvement, with one representative approach being Squeeze-and-Excitation Networks [24]. More recently, other studies [25,26] have also enabled networks to achieve better performance. These traditional attention mechanisms with image objects have better local perception in feature extraction and can fuse local information in deeper operations to generate global features. Attention mechanisms have also been used with great success in various tasks in computer vision, such as: image recognition, target detection, semantic segmentation, action recognition, image generation, 3D vision, etc.

In the field of deep learning, models often need to accept and process a large amount of data, however, at some specific moments, only a small amount of certain data is important, and at such times, it is necessary to focus our attention on these important data. The attention mechanism can be used as a resource allocation scheme, which is the main means to solve the information overload problem. It allows for the processing of more important information with limited computational resources in cases of limited computational power.

## 3. Method

### 3.1. Attention Fusion Module

In traditional channel attention, the attention is only focused on the importance of different channels and is not sensitive to positional information [27]. Unlike channel attention, spatial attention focuses on "where" the information part is, which is complementary to channel attention [25]. Because not all regions in the feature map are equally important in their contribution to the tool wear prediction task, only task-related regions are of concern. Spatial attention is about finding parts of the network that are relevant to the prediction task for processing. While in the input multi-scale feature information, features on different time scales correspond to different tool wear degradation information. If the multi-scale feature information extracted from the input sensor data is directly used to predict tool wear, it may lead to suboptimal results due to the location irrelevance. Although channel attentions have significant effects in improving model performance, they usually ignore location information, which is important for the generation of spatially selective attention maps [28]. Therefore, to solve the problem of unsatisfactory results due to irrelevant location information, embedding location information into channel attention is considered. In the attention fusion module of Figure 2, the importance of the data is encoded from the horizontal and vertical directions respectively, which is quite different from the squeezing operation in the channel attention method which produces a single feature vector. It can capture long-range dependencies in one direction and retain precise location information in the other direction, which helps the network learn degenerate features more accurately.

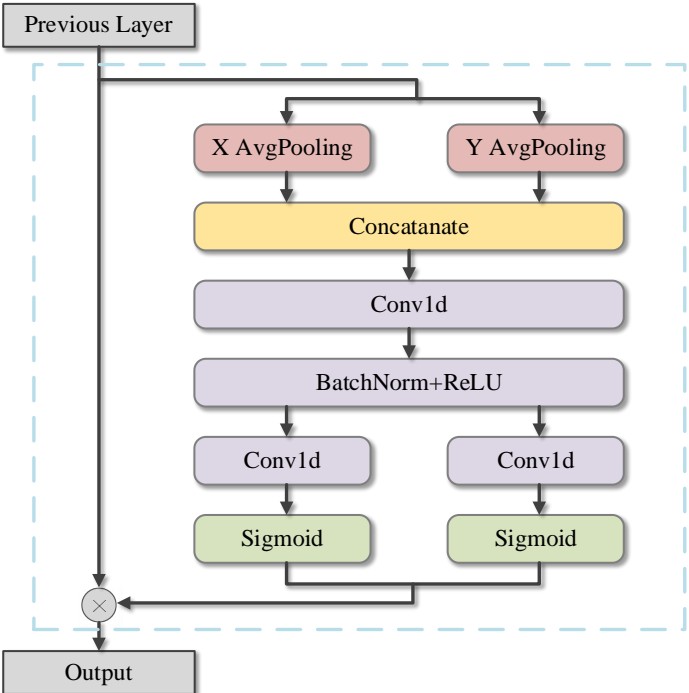

**Figure 2.** The structure of the attention fusion module.

In channel attention, the global pooling operation is usually used to compress each channel to obtain inter-channel correlation information, but this operation obviously ignores the location information between unused channels. To enable the attention module to retain the location information, the global pooling operation is first compressed along the temporal and spatial dimensions, respectively [29], as shown in Equations (3) and (4).

$$z_c = \frac{1}{L} \sum_{i=1}^{L} x_c(i), \tag{3}$$

$$z_x = \frac{1}{c} \sum_{j=1}^{c} x_l(j), \tag{4}$$

where $c$ denotes the number of channels, $L$ denotes the length of the sequence, $z_c$ denotes the compressed output of the $c$th channel and $z_x$ denotes the compressed output of the $x$th data position.

The second step is to aggregate the features of two dimensions and further use the activation function to quantify the importance of different convolution channels. The activation function can use a two-layer fully connected network.

$$f = \sigma_2(W_2 \sigma_1(W_1([z_c, z_x]))), \tag{5}$$

where $[\cdot]$ denotes the aggregation operation along the spatial direction, $\sigma_1$ and $\sigma_2$ denote the activation functions, $W_1$ and $W_2$ denote the corresponding weights, $f \in \mathbb{R}^{c+L}$, and then $f$ is split into channel weights $f_c \in \mathbb{R}^c$ and position weights $f_l \in \mathbb{R}^L$. Finally, the output of the attention module can be written as:

$$y(i) = x(i) \times f_i^c \times f_i^l, \tag{6}$$

### 3.2. The Multi-Scale Convolutional Network with Attention Fusion

The structure of the multi-scale convolutional network with attention fusion is shown in Figure 3. The model mainly consists of a multi-scale convolution module, attention fusion module, residual connection, and fully connected layers.

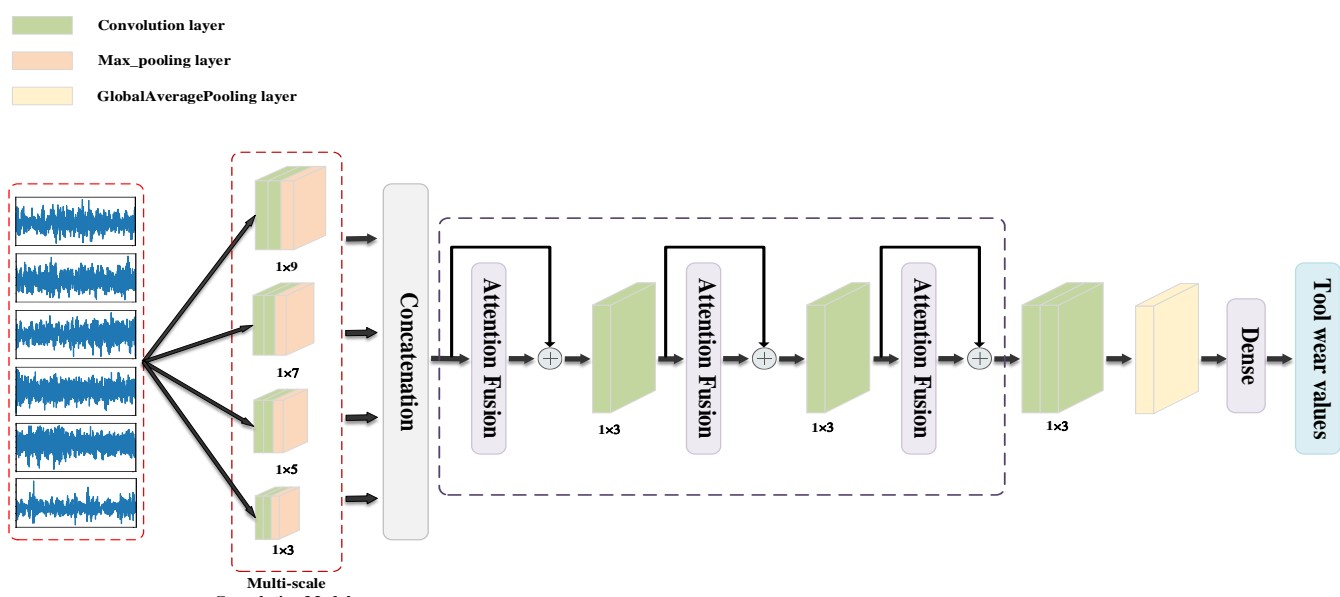

**Figure 3.** Network structure diagram of tool wear prediction.

In the multi-scale convolution module, four different convolution kernels of $1 \times 3$, $1 \times 5$, $1 \times 7$, and $1 \times 9$ are used for feature extraction. Furthermore, in the multi-scale convolution module in Figure 3, we use four graphs with the same shape but different sizes to represent the four convolution kernels, respectively.

Firstly, the multi-scale convolutional network with attention fusion takes different types of preprocessed monitoring sensor data as network input. Secondly, the multi-scale convolutional module learns the multi-sensor data to obtain the tool wear degradation information on different time scales from different types of monitoring data. At the same time, the network combines the attention fusion module to filter out the features extracted from the multi-scale convolution module that are strongly correlated with the tool wear degradation information, to learn the key degradation information exhibited by different sensors and reduce the influence of irrelevant and redundant information. Then, the network can retain the original feature information extracted from multi-sensor data and fuse the key degradation information selected by attention through the residual connection. The model can obtain richer degraded feature information through the attention mechanism with residual connections and reduce the gradient disappearance or gradient explosion problem caused by the deepening of the network. Finally, a fully connected layer is used to predict the tool wear value.

In Figure 3, concatenation indicates that the wear information extracted by the multi-scale convolution module is combined according to the dimension of its feature matrix, $\oplus$ indicates the addition operation of the corresponding feature vector, and the Dense indicates the fully connected layer.

### 3.3. The Flowchart of the Proposed Method

Firstly, all samples are divided into a training set, validation set and test set according to the ratio of 6:2:2. The training set and validation set are used to train the model and adjust the model parameters to determine the current optimal model, and the test set is used to evaluate the model performance The model uses the method of early stop to end the training, the number of early stop steps is set to 16 steps, and the evaluation function selects the mean square error loss, and its calculation is shown in formula (7).

$$E_{loss} = \frac{1}{n} \sum_{i=1}^{n} (y_i - \hat{y}_i)^2, \tag{7}$$

where $y_i$ and $\hat{y}_i$ denote the predicted and true values of the model, respectively, $n$ denotes the total amount of data in the validation set, and $E_{loss}$ denotes the value of the loss function in the validation set. Figure 4 shows the flow chart of the method proposed in this paper. The specific steps are as follows:

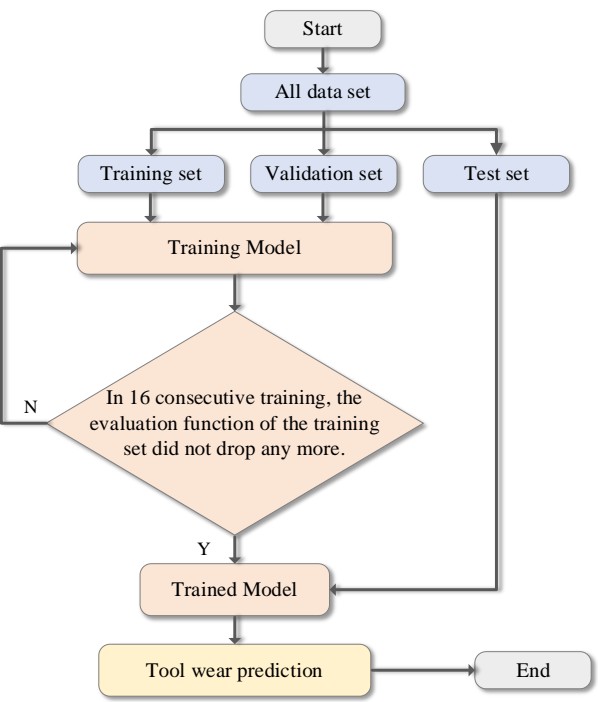

**Figure 4.** The flowchart of the proposed method.

(1)    The dataset obtained after pre-processing is divided into a training set, a validation set, and a test set.
(2)    The training set data is put into the model for training, and the validation set data is used to calculate the loss and gradient. The model updates parameters through gradient descent and determines whether the model needs to continue training according to the learning strategy.
(3)    After the model training is completed, the test set is input into the trained model to evaluate the performance of the model, thereby realizing tool wear prediction.

## 4. Experiment

### 4.1. Data Description

The tool wear data set from the American PHM Association in 2010 was used for this study [30]. The model of the high-speed CNC machine used in the experiment is Röders Tech RFM760 from Soltau, Germany. The workpiece material used in the machine tool processing test is stainless steel (HRC52), and the test tool is a 6 mm ball-end carbide milling cutter. The experimental processing parameters are shown in Table 1.

**Table 1.** Experimental processing parameters table.

| Spindle Speed (r/min) | Feed Rate (mm/min) | Radial Depth of Cut (mm) | Axial Depth of Cut (mm) | Sampling Frequency (kHz) |
|---|---|---|---|---|
| 10,400 | 1555 | 0.125 | 0.2 | 50 |

In the experiment, an acceleration sensor, dynamometer and acoustic emission sensor were used to collect vibration signal, force signal and acoustic emission signal in the machining process of the machine tool, and the data acquisition system is shown in Figure 5. The experiment uses a Kistler three-way force measuring instrument to collect the cutting

force in the X, Y and Z directions, and its installation position is between the workpiece and the worktable. Three Kistler piezoelectric accelerometers were used to collect the vibration signals of the machine tool in the X, Y and Z directions of the cutting process, and Kistler acoustic emission sensors were used to collect the acoustic emission signals generated during the cutting process. The sensor signals were sampled at a sampling frequency of 50 kHz and the acquired data was saved every 15 s, including approximately 200,000 signals. In each experiment, the tool was used to continuously cut the bevel of the workpiece to machine a complete bevel. After cutting one surface, the tool was removed and the flank wear values of the three cutting edges were measured using a microscope.

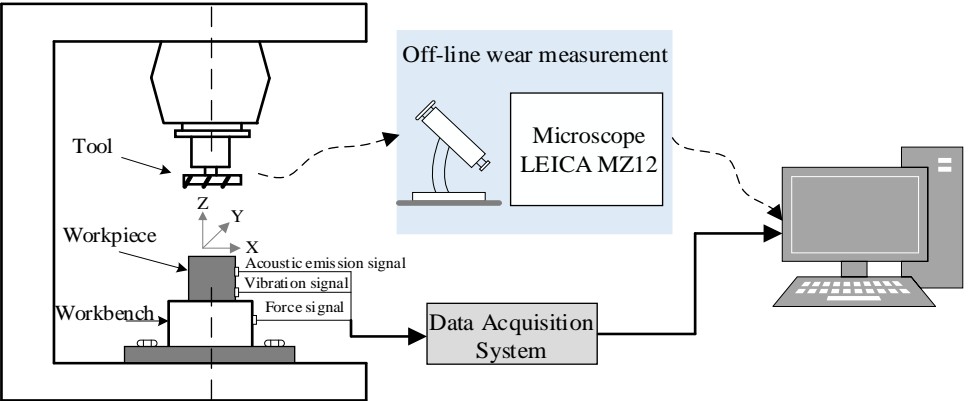

**Figure 5.** Experimental platform.

### 4.2. Data Processing

The public dataset contains a total of six tools with full life cycle data. Each tool has a total of 315 milling data collected, of which only C1, C4 and C6 contain labels, so only C1, C4 and C6 data are used as the database for the study in this paper. The maximum values of the three blade wear data for each tool C1, C4 and C6 were taken as label data. For the high sampling frequency of the original data and the possible loss of some information after too much down-sampling of the data, this paper performs outlier preprocessing and normalized preprocessing on the original vibration signal and cutting force data of each tool, and then performs equal interval sampling to obtain samples of dimension (2048, 6), 315 samples for each tool, and a total of 945 samples for the three tools. Among them, 80% of the samples are randomly selected as the training and validation sets to train and update the model, and 20% of the data are used as the test set to test the performance of the model.

### 4.3. Parameter Settings

In the training process of the model, it is also necessary to set some hyperparameters, such as the learning rate, the number of iterations, and the input data volume of each batch. The setting of these parameters has a great impact on the performance of the model. In this paper, the learning rate is set as 0.001 at the beginning. When the evaluation function value of the training set does not decrease in 8 consecutive trainings, the learning rate is reduced to 0.1 times of the original. The input data volume of each batch is 24 and the maximum number of iterations is 300. During training, we use the early stopping method to prevent overfitting. We set the number of early stopping steps to 16, that is, the training is ended when the value of the evaluation function no longer decreases during 16 consecutive training sessions, or when the evaluation function has continued to change, and training is stopped when the number of training times reaches the set 300 times to ensure that all models have converged. The loss function of the training set is MSE.

### 4.4. Evaluation Indicators

In this paper, the mean square error (MSE), mean absolute value error (MAE), determination coefficient ($R^2$) and mean absolute percentage error (MAPE) are used to evaluate

the overall performance of the model by comparing the four evaluation indexes. The evaluation index calculation formula is as follows:

$$MSE = \frac{1}{n} \sum_{i=1}^{n} (y_i - \hat{y}_i)^2, \tag{8}$$

$$MAE = \frac{1}{n} \sum_{i=1}^{n} |y_i - \hat{y}_i|, \tag{9}$$

$$R^2 = 1 - \sum_{i=1}^{n} (y_i - \hat{y}_i)^2 / \sum_{i=1}^{n} (y_i - \overline{y})^2, \tag{10}$$

$$MAPE = \sum_{i=1}^{n} \left| \frac{y_i - \hat{y}_i}{y_i} \right| \times \frac{100}{n}, \tag{11}$$

In the formula, $y_i$ represents the $i$th actual value, $\hat{y}_i$ represents the $i$th predicted value, and $\overline{y}$ represents the average of the $n$ actual values.

### 4.5. Experimental Results and Analysis

In order to verify the effectiveness of the model proposed in this paper and compare the superiority of other models, this paper compares the proposed model with the MSCNN model, and MSCNN with different attention mechanisms, including the squeeze-and-excitation attention (SE) [24], efficient lightweight channel attention (ELCA) [31], mixed self-attention (MAS) [32] and Convolutional Block Attention Module (CBAM) [25]. The attention fusion module is not used in MSCNN, and other settings are consistent with the model proposed in this paper. Among the different attention mechanism models, apart from the fact that the models use different attention mechanisms, the rest of the structure is consistent with the proposed model. The five models are trained and predicted on the test set, and the prediction results are shown in Table 2. The performance of the method proposed in this paper, MSCNN, MSCNN + SE and other models on tool wear prediction under the dataset are shown in Figures 6–11. In addition, we also add noise to the sensor signal to see which signals contribute to the tool wear prediction and how it affects the accuracy of the model. The experimental results are shown in Tables 3–5.

**Table 2.** Experimental processing parameters table.

| Model | Evaluation Indicators ± STD | | | |
|---|---|---|---|---|
| | **MAE** | **MSE** | **MAPE/%** | **$R^2$** |
| MSCNN | $5.65 \pm 2.42$ | $49.13 \pm 31.68$ | $4.94 \pm 0.19$ | $0.968 \pm 0.02$ |
| MSCNN + SE | $5.39 \pm 2.06$ | $51.56 \pm 42.06$ | $4.76 \pm 0.17$ | $0.955 \pm 0.05$ |
| MSCNN + ELCA | $5.44 \pm 1.33$ | $50.71 \pm 22.98$ | $4.84 \pm 0.11$ | $0.969 \pm 0.01$ |
| MSCNN + MAS | $5.59 \pm 0.59$ | $45.92 \pm 13.39$ | $4.98 \pm 0.06$ | $0.967 \pm 0.01$ |
| MSCNN + CBAM | $11.53 \pm 6.52$ | $223.01 \pm 217.29$ | $9.48 \pm 0.05$ | $0.873 \pm 0.12$ |
| The proposed model | $4.29 \pm 1.11$ | $34.00 \pm 13.91$ | $3.70 \pm 0.09$ | $0.975 \pm 0.01$ |

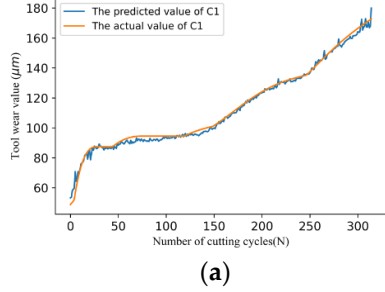
(**a**)

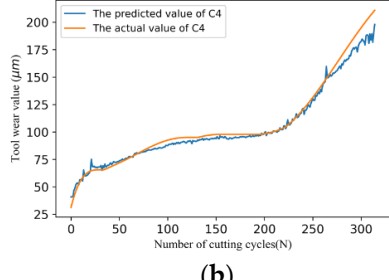
(**b**)

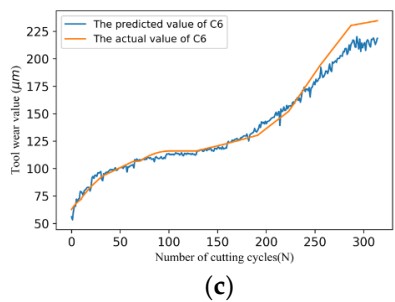
(**c**)

**Figure 6.** The prediction results of the model proposed in this paper on datasets. (**a**) C1, (**b**) C4, and (**c**) C6. $P_{MAE} = 4.29$, $P_{MSE} = 34.00$.

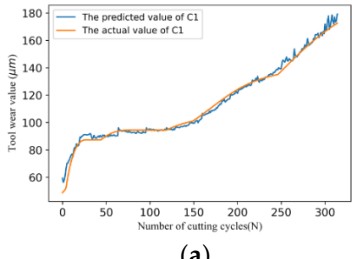
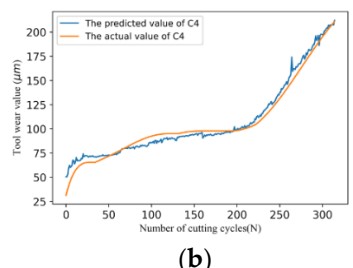
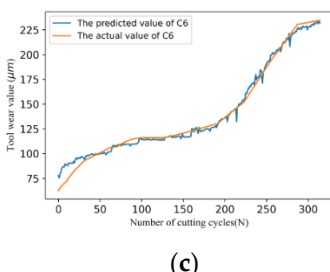

(**a**) (**b**) (**c**)

**Figure 7.** Prediction result of MSCNN model on datasets. (**a**) C1, (**b**) C4, and (**c**) C6. $P_{MAE} = 5.65$, $P_{MSE} = 49.13$.

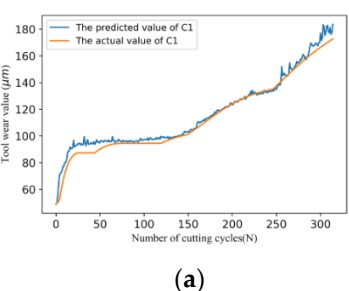
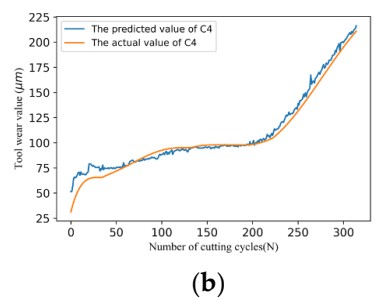
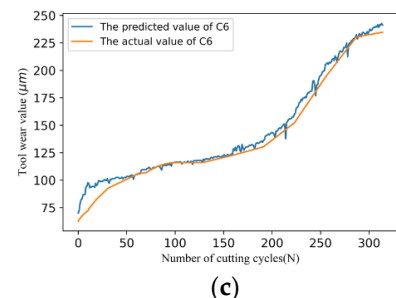

(**a**) (**b**) (**c**)

**Figure 8.** Prediction results of MSCNN + SE model on datasets. (**a**) C1, (**b**) C4, and (**c**) C6. $P_{MAE} = 5.39$, $P_{MSE} = 51.56$.

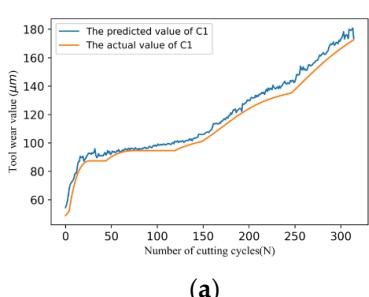
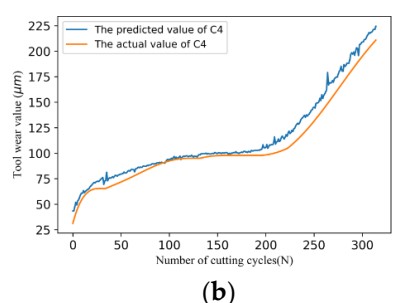
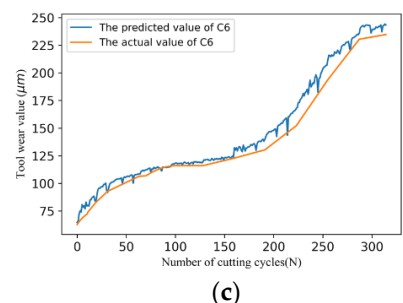

(**a**) (**b**) (**c**)

**Figure 9.** Prediction results of MSCNN + ELCA model on datasets. (**a**) C1, (**b**) C4, and (**c**) C6. $P_{MAE} = 5.44$, $P_{MSE} = 50.71$.

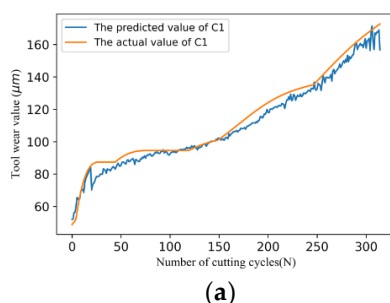
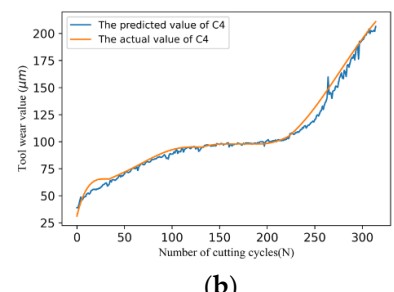
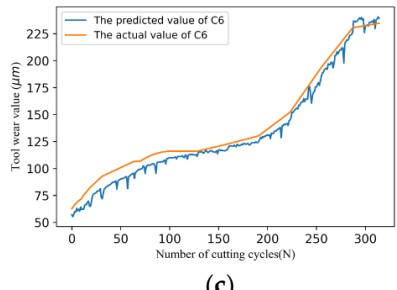

(**a**) (**b**) (**c**)

**Figure 10.** Prediction results of MSCNN + MAS model on datasets. (**a**) C1, (**b**) C4, and (**c**) C6.$P_{MAE} = 5.59$, $P_{MSE} = 45.92$.

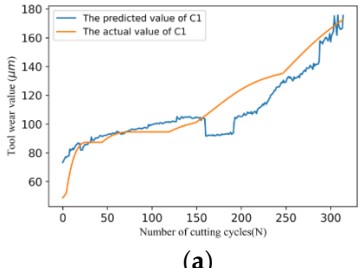 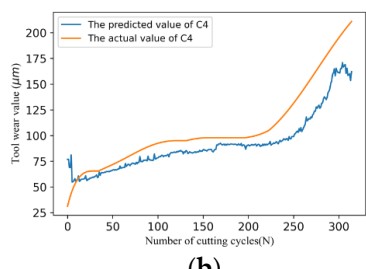 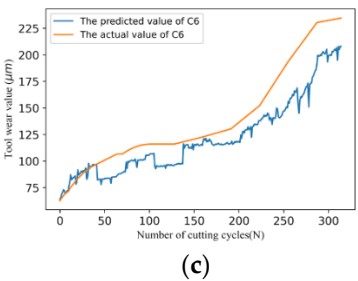

(**a**) (**b**) (**c**)

**Figure 11.** Prediction results of MSCNN + CBAM model on datasets. (**a**) C1, (**b**) C4, and (**c**) C6. $P_{MAE} = 11.53$, $P_{MSE} = 223.01$.

**Table 3.** Experimental results of adding white Gaussian noise to cutting force signals.

| Model | MAE | MSE | MAPE/% |
|---|---|---|---|
| MSCNN + ELCA | 49.57 | 3037.93 | 72.71 |
| MSCNN + MAS | 35.29 | 1815.92 | 40.17 |
| MSCNN + CBAM | 75.73 | 6434.62 | 194.94 |
| The proposed model | 30.08 | 1783.13 | 30.24 |

**Table 4.** Experimental results of adding white Gaussian noise to vibration signals.

| Model | MAE | MSE | MAPE/% |
|---|---|---|---|
| MSCNN + ELCA | 19.53 | 647.84 | 15.29 |
| MSCNN + MAS | 18.51 | 659.42 | 15.03 |
| MSCNN + CBAM | 22.34 | 810.00 | 18.16 |
| The proposed model | 17.04 | 527.03 | 14.21 |

**Table 5.** Experimental results of adding white Gaussian noise to all signals.

| Model | MAE | MSE | MAPE/% |
|---|---|---|---|
| MSCNN + ELCA | 13.96 | 234.46 | 15.64 |
| MSCNN + MAS | 22.79 | 574.39 | 24.99 |
| MSCNN + CBAM | 20.59 | 597.66 | 15.83 |
| The proposed model | 6.41 | 84,045 | 6.01 |

From the experimental results in Figures 6–11, it can be found that the model of the proposed method in this paper can well reflect the variation of the real tool wear values. At the same time, according to its evaluation indexes MAE and MSE, it can be seen that the error between the predicted value of the model and the real value of tool wear is the smallest among several comparative models. The result verifies the effectiveness and superiority of the proposed tool wear prediction model based on the multi-scale convolutional attention fusion. In addition, it also shows that the model has a large improvement in the accuracy of tool wear prediction.

It can be seen from Table 2 that in addition to the methods proposed in this paper, MSCNN has achieved relatively good results in each index, which indicates that the multi-scale network can extract multi-sensor information, and the network has good tool wear prediction performance. Through the improvement of the SE attention mechanism and comparing it with several other comparative models in Table 2, it can be seen that the model has achieved good results on MAE, indicating that SE can improve the accuracy of the model output, and reduce the mean of the absolute error between the predicted value and the true value. Through the improvement of the MAS, the model achieved the best results on MSE, indicating that MAS can well reduce the degree of dispersion of the predicted value, and the deviation between the actual value and the predicted value fluctuates less. Through the improvement of the ELCA attention mechanism, the model achieves the best

results on MAPE and R2, indicating that the ELCA attention mechanism can achieve a better overall prediction effect and effectively reduce the overall difference between the predicted value and the actual value. In addition, by comparing $R^2$, ELCA has a better fitting effect.

In addition, it can be seen from Table 2 that the method proposed in this paper has achieved the best results in the four performance indicators. Through comparative experiments, the MAE index is lower than the lowest value of other methods from 5.39 to 4.29; the MSE indicator is lower than the lowest value of other methods from 45.92 to 34.00; the MAPE indicator is lower than the lowest value of other methods from 4.76% to 3.70%; the $R^2$ indicator was improved to 0.975 compared to the highest value of 0.969 by other methods.

Tables 3–5 add white Gaussian noise to the cutting force signal, vibration signal and all signals, respectively. According to Tables 3–5, it can be seen that when white Gaussian noise is added to the cutting force signal alone, the evaluation function indicators of these comparison models all change the most, indicating that the cutting force signal will help better predict tool wear. In addition, according to the experimental results of these three noise additions, the method proposed in this paper has achieved good results. According to Tables 2 and 5, it can be seen that when white Gaussian noise is added to the vibration force signal and cutting force signal at the same time, the MAE of the method proposed in this paper has little difference in value compared with the MAE without noise, indicating that the method is very good at improving the accuracy of the model output, and has good performance in reducing the average value of the absolute error between the predicted value and the true value. Compared with several other models, it can be seen that the method proposed in this paper has achieved better results.

The above results show that the attention fusion module proposed in this paper can effectively fuse the feature information of different scales extracted by the multi-scale convolution module, so that the model can obtain the feature information strongly related to tool wear. At the same time, it verifies the effectiveness and superiority of the model proposed in this paper in tool wear prediction.

## 5. Conclusions

In this paper, a tool wear prediction method based on a multi-scale convolution network with attention fusion is proposed. Firstly, in order to effectively extract the tool degradation information from the monitoring data, a multi-scale convolution module is constructed. Then, considering that the tool wear information obtained by different sensor information through different convolution kernels is different, an attention fusion module is introduced to fuse multi-scale feature information. Compared with some existing attention methods, the experimental results show that the model proposed in this paper can more accurately predict the tool wear value. In future studies, we may carry out further work on the interpretability of the attention mechanism weights linked back to the original data.

**Author Contributions:** Conceptualization, Q.H. and H.H.; methodology, Y.H. and H.H.; software, D.W. and H.H.; validation, Y.Z. and Q.H.; formal analysis, D.W. and Y.H.; investigation, D.W. and H.H.; resources, H.H. and Y.Z.; data curation, Q.H. and Y.H.; writing—original draft preparation, D.W. and H.H.; writing—review and editing, Q.H. and Y.H. All authors have read and agreed to the published version of the manuscript.

**Funding:** This research was funded by the National Key R & D Program of China under Grant 2021YFB3301000, the Natural Science Foundation of Chongqing (cstc2021jcyj-msxmX0556), the Chongqing Postdoctoral Science Foundation (cstc2021jcyj-bshX0094), Chongqing Municipal Education Commission Science and Technology Research Project (KJQN202000611), Chongqing Yubei District Big Data Intelligent Technology Special Key Project (2020-02), the Venture & Innovation Support Program for Chongqing Overseas Returnees (No. cx2021026), and the funding for postdoctoral research projects of Chongqing (No. 2021XM3091).

**Institutional Review Board Statement:** Not applicable.

**Informed Consent Statement:** Not applicable.

**Data Availability Statement:** The article contains the data which are also available from the corresponding authors upon reasonable request.

**Conflicts of Interest:** The authors declare no conflict of interest.

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
