# Peer review of "Tool Wear Prediction Based on a Multi-Scale Convolutional Neural Network with Attention Fusion"

_information, doi:10.3390/info13100504_

Round 1
Reviewer 1 Report
The paper presents a tool wear prediction method based on multi-scale convolutional neural network with attention fusion, which fuses the tool wear degradation information collected by different types of sensors. The paper is organized well, but there are still some major issues that need to be resolved.
1. The abstract should be revised, as it does not adequately describe the research questions (challenges) addressed by the proposed method in relation to the research background in the field of tool wear prediction methods based on CNNs
2. In line 67, why can the network not efficiently select task-relevant features in the face of different sensor data? This is supposed to be the main challenge addressed in this paper, and the authors are advised to explain the reasons in more detail.
3. The structure of the paper can be better, especially for Section 2 and Section 3. The authors are suggested to add a " Related Work" section to include relevant research on multi-scale convolutional neural network and attention fusion module.
4. In fact, there has been a lot of research on attentional mechanisms in the cv field. For example, CBAM is a popular module combining the channel attention mechanism and the spatial attention mechanism. What are the advantages of the proposed Attention Fusion Module in this paper compared to these modules?
5. The network structure in Figure 3 is confusing. The authors claim to use multiple different size convolutional kernels but do not provide details on which kernel size is specifically used for each convolutional layer. Moreover, the description of the feature map shape is missing.
6. In Section 4, authors are suggested to add a comparison of the proposed method with the state-of-the-art.
7. Confusion is created with the use of “multiscale convolution” and “multi-scale” in the paper. The authors may consider unifying the expression.
8. The experimental results and analysis of the paper should be enhanced so as to describe the advantages of the proposed methods. For example, I suggest the authors provide loss curves for analysis.
Author Response
The authors deeply thank the reviewers for reviewing this paper with great interest and feedback. Please see the attachment.

Reviewer 2 Report
The paper presents a new deep learning architecture for the prediction of tool wear based on multi-scale features and attention mechanisms.
The paper is in general well written. There are a few points that need to be addressed by the authors to improve the quality of the work:
1. There are several statements on the paper that are not very well explained or not enough brackground is provided to sustain the claims. Further details need to be added:
Statement 1 (Page 2, line 67) - "However, the network cannot effectively select task-related features in the face of different sensor data, the accuracy of the tool wear prediction by the multi-scale network
cannot be further improved".
From the way it is written, it seems the authors say that without no attention mechanisms, then multio-scale approaches cannot be improved any further. To be able to
afirm this, the authors would have had to try every single possible approach/architecture and deep learning mechanisms to be confident this is true. Unless I am not interpreting correctly
the sentence and a rewording would then be advised.
Statement 2 (Page 3, line 116) - If the multi-scale feature information extracted from the input sensor data is directly used to predict tool wear, it may lead to suboptimal results due
to the location irrelevance.
This sentence references paper 25. Paper 25 is not a tool wear paper, but a paper to reduce the computational complexity of using a multi-scale deep learning architecture. How can the authors
make this conclusions for tool wear from paper 25? To what extent can the sensor signals change to affect the accuracy of a tool wear prediction approach? What does suboptimal means for tool wear?
and how is the result achieved by the authors approach optimal? There can always be another approach that obtains a smaller MAE.
2. Data pre-processing section is missing several details for anyone to try and reproduce this work:
*How large is your data set? if signals are sampled at 50Khz continuously for various seconds for 315 cuts how did you sample your data? how large is actually each input signal in each sample?
*Did you took smaller segments from the signal obtained in each cut?
*If so, How many total samples you got after cutting your signal in segments?
*Is the signal normalised?
*It is not explained in the paper that the data set comes with measurements of the different tool edges, which of these measurements did you take as the tool wear value? did you average it?
* did you mix data of all three cutters? or did you use some cutters for training and some for test?
3. A figure of the tool wear curve would be helpful to support the data description section
4. 16 consecutive trainings seems to be an arbitrary value, the same with 300. Was this determined experimentally?
5. A weak point on this work is that a good explanation/analysis is missing on to how is the spatial relationship key in tool wear to increase the accuracy. What is it about the features in
the sensor signal that makes it an important factor to incorporate these attention mechanisms for tool wear prediction. Some signal analysis would be good to understand better why this is the case.
Also, it is said that attention is a good thing to have in a deep learning model because it helps with interpretability. Understanding better how the model is making the predictions.
A section on interpreting these results would improve the quality of this paper. What signals are then contributing more to a better estimation of tool wear, and what particular aspects
of the signal contribute to a more accurate result? In the same way, what type of noise in the signal the model is able to ignore using attention that allows it to have a better accuracy?
An additional experiment is advised, making a change in the signal, some noise, and see how it could affect or not affect the accuracy of the model . this could help understand the sensitivity of the method and how it is alleviated by the attention mechanism.
6. A plot of the predicted versus the real value would be good to have. Did you use all cutters to train? you could use one cutter to test to be able to plot results in a typical tool wear curve.
Author Response

(The authors gave the same response as above.)

Round 2
Reviewer 1 Report
The paper proposed a novel tool wear prediction framework using multi-scale CNN and attention fusion, which is well illustrated. Thanks to the authors for their effort in improving their manuscript. The authors satisfied reviewer’s comments. In addition, related work on attention mechanisms in the CV domain and their limitations are suggested to be added in 2.2.
Author Response

(The authors gave the same response as above.)

Reviewer 2 Report
The authors have addressed all comments. It is however still unclear how the location attention improves the result, or what role this plays in the accuracy and so to help the reader understand this more analysis and explanation is required. Can you link location back to an area of the original input? Does the relevant location and features change as the tool wears more? Visualisation of the input signal and also where the attention mechanisms focused after feeding the input to the network with a heat map perhaps are suggested. Showing these perhaps with samples at different stages to tool wear.
Please add this analysis for completeness.
Author Response

(The authors gave the same response as above.)
